# Adherence of *Trichomonas vaginalis* to SiHa Cells is Inhibited by Diphenyleneiodonium

**DOI:** 10.3390/microorganisms8101570

**Published:** 2020-10-13

**Authors:** Yeeun Kim, Young Ha Lee, In-Wook Choi, Bu Yeon Heo, Ju-Gyeong Kang, Jae-Min Yuk, Guang-Ho Cha, Eun-Kyeong Jo, Jaeyul Kwon

**Affiliations:** 1Department of Medical Science, College of Medicine, Chungnam National University, 266 Munhwa-ro, Jung-gu, Daejeon 35015, Korea; dpdms1189@hanmail.net (Y.K.); yhalee@cnu.ac.kr (Y.H.L.); choi76@cnu.ac.kr (I.-W.C.); xeyk1603@naver.com (B.Y.H.); yjaemin0@cnu.ac.kr (J.-M.Y.); gcha@cnu.ac.kr (G.-H.C.); hayoungj@cnu.ac.kr (E.-K.J.); 2Department of Infection Biology, College of Medicine, Chungnam National University, Daejeon 35015, Korea; 3Cardiovascular Branch, National Heart, Lung, and Blood Institute, NIH, Bethesda, MD 20892, USA; kangju7@kaist.ac.kr; 4Department of Biological Sciences, Korea Advanced Institute of Science and Technology, Daejeon 34141, Korea; 5Infection Control Convergence Research Center, College of Medicine, Chungnam National University, Daejeon 35015, Korea; 6Department of Microbiology, Collage of Medicine, Chungnam National University, Daejeon 35015, Korea; 7Department of Medical Education, College of Medicine, Chungnam National University, Daejeon 35015, Korea; 8Translational Immunology Institute, Chungnam National University, Daejeon 35015, Korea

**Keywords:** *Trichomonas vaginalis*, adherence, SiHa cells, diphenyleneiodonium (DPI), parasite, infection

## Abstract

Microbial adhesion is critical for parasitic infection and colonization of host cells. To study the host–parasite interaction in vitro, we established a flow cytometry-based assay to measure the adherence of *Trichomonas vaginalis* to epithelial cell line SiHa. SiHa cells and *T. vaginalis* were detected as clearly separated, quantifiable populations by flow cytometry. We found that *T. vaginalis* attached to SiHa cells as early as 30 min after infection and the binding remained stable up to several hours, allowing for analysis of drug treatment efficacy. Importantly, NADPH oxidase inhibitor DPI treatment induced the detachment of *T. vaginalis* from SiHa cells in a dose-dependent manner without affecting host cell viability. Thus, this study may provide an understanding for the potential development of therapies against *T. vaginalis* and other parasite infections.

## 1. Introduction

Trichomoniasis is a sexually transmitted disease (STD) caused by infection of the urogenital tract by the flagellate protozoan parasite *Trichomonas vaginalis* [1,2,3]. Trichomoniasis in pregnant women may lead to premature membrane rupture, preterm delivery, and low birth weight [4]. It has also been associated with atypical pelvic inflammatory disease [5], infertility [6], predisposition to invasive cervical cancer [7], and increased susceptibility to HIV infection [8]. While readily curable with antibiotics, most infections are asymptomatic and left untreated, thereby increasing the chances of its transmission and significant health consequences. Indeed, in 2008 the WHO estimated an incidence of 276 million new trichomoniasis cases per year, making it the most common nonviral STD in the world [9]. Despite its prevalence and serious health effects, trichomoniasis has received inadequate attention by health professionals. Of particular concern, antibiotic-resistant strains of *T. vaginalis* may be emerging. The exact prevalence of nitroimidazole-resistant *T. vaginalis* infections is unknown, largely due to the lack of standardized antibiotic susceptibility tests or surveillance systems to detect treatment failures due to resistance [10]. Therefore, considerable efforts to establish a standardized antibiotic susceptibility test and identify new anti-*T. vaginalis* drugs are highly warranted [11].

Adhesion of *Trichomonas vaginalis* to host mucosal cells is considered to be an initial and essential step for its infection [12]. The adhesive properties of *T. vaginalis* are intimately related to its virulent characteristics and several classes of molecules have been found to be critical to the *T. vaginalis*–host interaction, including *T. vaginalis* lipophosphoglycan (TvLPG), adhesins, BspA (bacteroides surface protein A)-like, and cadherin-like protein [13,14,15,16,17]. Upon contact with primary vaginal epithelial cells, *T. vaginalis* has been shown to undergo rapid actin cytoskeleton reorganization to transition from a flagellate to an elongated amoeboid shape with pseudopodia [18,19]. Several studies have highlighted signal transduction pathways that may underly these cytoskeletal dynamics [18,20,21]. However, although reactive oxygen species (ROS) generated by NADPH oxidases have been shown to regulate actin dynamics and adhesion in several other systems [22,23,24], the role of ROS in *T. vaginalis* morphogenesis has yet to be addressed.

Understanding parasite adhesion is critical to gaining insight on how host cells are infected and colonized. In this study, we established a flow cytometry-based method to examine the properties of *T. vaginalis* adhesion on the cervix carcinoma cell line, SiHa cells. Using antioxidant compounds such as N-acetyl-L-cysteine (NAC) and NADPH oxidase inhibitor diphenyleneiodonium (DPI), we determined whether ROS production was involved in the adhesion process during *T. vaginalis* infection. We found that, DPI, but not NAC, was involved in blockade of parasite adherence to host cells, suggesting that DPI-targetable ROS generation in *T. vaginalis* contributes to the pathogenic adherence process.

## 2. Materials and Methods

### 2.1. Host Cell Culture

Human cervical epithelial cancer cell lines (SiHa cells) were obtained from the American Type Culture Collection (ATCC, Manassas, VA, USA). SiHa cells were maintained under 5% CO_2_ at 37 °C in Dulbecco’s modified Eagle’s medium (DMEM) with 10% heat-inactivated fetal bovine serum (FBS, Gibco BRL, Grand Island, NY, USA) and 1 × antibiotic–antimycotic (Anti-Anti, Gibco BRL).

### 2.2. T. vaginalis Culture

*T. vaginalis* T106 was provided by Prof. J. K. Alderete (University of Texas, Health Science Center, TX, USA) and cultured in TYM medium consisting of 20% (*w*/*v*) trypticase peptone, 10% (*w*/*v*) yeast extract, 5% (*w*/*v*) Moltose monodydrate, 1% (*w*/*v*) L-cystein hydrochloride, 1% (*w*/*v*) ascorbic acid, 1% (*w*/*v*) K_2_HPO_4_, 1% (*w*/*v*) KH_2_PO_4_ (pH 6.2) with 10% heat-inactivated horse serum (Sigma-Aldrich, St. Louis, MO, USA) and penicillin-streptomycin (Gibco BRL) and incubated under 5% CO_2_ at 37 °C SiHa cells were infected with live *T. vaginalis* at multiplicities of infection (MOI) of 1 in the mixed-medium (DMEM without Antibiotic: TYM = 2:1) as previously described [25].

### 2.3. Flow Cytometry Analysis

Forward scatter (FSC) and side scatter (SSC) were measured for gating and FSC-A and FSC-H were used for the detection of single cells. Typically, SiHa cells and bound *T. vaginalis* were acquired at 10,000 events. To assess SiHa cell and *T. vaginalis* viability, cell pellets were resuspended in flow cytometry buffer containing 5 μg/mL propidium iodide (PI) (Sigma-Aldrich) and analyzed on a FACSCalibur (BD Immunocytometry Systems, San Jose, CA, USA) as previously described [26]. Only live SiHa cells (PI-negative) were used for ROS detection via CM-H_2_DCFDA. Flow cytometry was analyzed by BD Canto II Flow cytometer (BD Biosciences, San Jose, CA, USA) and FlowJo software (Treestar, Ashland, OR, USA).

### 2.4. Measurement of ROS Production

SiHa cells were pretreated with an antioxidant N-acetyl-L-cysteine (NAC, Sigma-Aldrich) or the NADPH oxidase inhibitor diphenyleneiodonium (DPI, Sigma-Aldrich) prior to or at the same time of *T. vaginalis* infection. The general oxidative stress indicator 5-(and-6)-chloromethyl-2′,7′-dichlorodihydrofluorescein diacetate, acetyl ester (CM-H_2_DCFDA, Life Technologies, Carlsbad, CA, USA) was used to detect reactive oxygen species (ROS) in live cells, as previously described [27]. Single endpoint measurements of ROS in SiHa and *T. vaginalis* cells were determined by adding DCFDA at 15 min before harvesting and washing the cells. Dye oxidation in cells treated with NAC or DPI was calculated as the percentage increase of mean channel fluorescence relative to that of untreated cells at each time point using the following equation: [(MCF_treat_ − MCF_untreat_)/MCF_untreat_] × 100(1)

### 2.5. Statistical Analysis

Experiments were repeated at least three times and expressed as the mean ± standard deviation (SD). *p*-values between groups were determined by two-tailed unpaired *t*-test or one-way ANOVA with Tukey’s post-test using GraphPad Prism (v7.02, GraphPad, San Diego, CA, USA). *p* < 0.05 was considered statistically significant.

## 3. Results

### 3.1. Flow Cytometric Analysis of Parasite–Host Cell Mixed Culture

In previous studies, fluorescent dye-loaded *T. vaginalis* had been incubated with host cells and monitored by microscopy or flow cytometry [28,29]. Here, we developed a quantitative flow cytometry assay to identify *T. vaginalis* and SiHa cells in a mixed population without the use of fluorescent dyes and separately assess the properties of each group. Cells from the cervix carcinoma cell line (SiHa) were infected with live *T. vaginalis* trophozoites, incubated for the time indicated, and washed with PBS to remove unbound cells. The remaining attached cells were trypsinized and analyzed by flow cytometry. Due to the remarkable differences in cell size, SiHa cells and *T. vaginalis* were clearly detected as separate populations in the FSC-A and SSC-A dot plot (Figure 1A). It is of note that *T. vaginalis* attached to SiHa cells as early as 30 min after infection and maintained attachment for up to three hours. In contrast, prolonged infection (over 18 h) lead to a reduction in SiHa host cells to around 20% of the total cell number and a concomitant increase in the percent population of *T. vaginalis* cells (Figure 1B–D). As identified by propidium iodide (PI) positive staining, the proportion of dead SiHa cells increased in infections greater than 18 h, indicating that prolonged infection of *T. vaginalis* is detrimental to host cells and in agreement with previous reports [25] (Figure 1E). Overall, our data suggest that *T. vaginalis* immediately adhere to SiHa cells and maintain attachment status, ultimately allowing *T. vaginalis* to induce host cell lysis.

### 3.2. DPI Reduces Adherence of T. vaginalis to SiHa Cells

It is known that ROS play important roles in infectious disease. ROS generated by NADPH oxidases has been shown to regulate cell adhesion, where antioxidant drugs were successfully used to modulate cell–cell interactions in various other systems [22,23,24]. Using our flow cytometry-based method, we sought to test the effect of NAC and DPI on the attachment of *T. vaginalis*. SiHa cells were treated with 10 mM NAC or 1 μM DPI for the last 2 h of the indicated *T. vaginalis* infection durations (Figure 2A). Interestingly, treatment with NADPH oxidase inhibitor DPI induced detachment of *T. vaginalis* from SiHa cells, whereas NAC, a widely used general antioxidant, did not affect the host–parasite interaction (Appendix A, Figure 2B,C). 

### 3.3. Antioxidant Pretreatment of SiHa Cells Does Not Affect T. vaginalis Adhesion

To test whether pretreating host cells with antioxidants could affect parasite binding, SiHa cells were treated with NAC or DPI for 1 h. After washing with PBS, pretreated SiHa cells were infected with *T. vaginalis* trophozoites for 30 min or 1 h and analyzed by flow cytometry (Figure 3A). Bound *T. vaginalis* levels were unchanged by pretreatment with NAC or DPI, suggesting that DPI affects the infection process rather than the host cell biology (Figure 3B,C). 

### 3.4. DPI Induces Detachment of T. vaginalis from SiHa Cells and Reduces ROS Generation

To find the optimal concentration of DPI that can induce *T. vaginalis* detachment, SiHa cells were infected with *T. vaginalis* for 3 h and incubated with various concentration of DPI for the last 2 h. To monitor ROS generation, 1 μM CM-H_2_DCFDA was added for the last 15 min. (Figure 4A). The levels of attached *T. vaginalis* were markedly decreased in a dose dependent manner (Figure 4B,C). On the other hand, DPI did not impact viability when given to either SiHa cells or *T. vaginalis* alone (Appendix A). Infected SiHa cells treated with DPI did not show statistically significant changes in relative DCFDA oxidation in a dose-dependent manner, whereas bound *T. vaginalis* showed significant reductions in DCFDA oxidation following 0.5 μM or 1 μM DPI treatment (Figure 4D). Taken together, DPI treatment induced the detachment of *T. vaginalis* from SiHa cells in a dose-dependent manner that may be associated with decreased ROS production in *T. vaginalis*.

### 3.5. Kinetics of DPI-Induced Detachment of T. vaginalis from SiHa Cells

To examine the kinetics of DPI treatment on *T. vaginalis* detachment, SiHa cells were infected with *T. vaginalis*, incubated with 0.5 μM of DPI for an additional 20–120 min, and loaded with CM-H_2_DCFDA for the last 15 min (Figure 5A). DPI treatment induced the detachment of *T. vaginalis* as early as 20–40 min and maintained its effect up to 2 h (Figure 5B,C). Similarly, DPI significantly reduced DCFDA oxidation in bound *T. vaginalis* at all time points examined (Figure 5D). Together, 0.2–0.5 μM DPI treatment for 20–40 min is sufficient to induce detachment of *T. vaginalis* following infection of SiHa cells.

## 4. Discussion

In this study, we reported a new effect of the well-known NADPH/NADH oxidase inhibitor DPI on the attachment of flagellate parasite *T. vaginalis* to a human cervical cancer cell line using flow cytometry analysis of mixed cell cultures. DPI treatment was able to rapidly induce the detachment of *T. vaginalis* from its parasitic adhesion on host SiHa cells. Given the concomitant reduction in DCFDA oxidation following DPI treatment, the effects of DPI may be associated with ROS production in *T. vaginalis*. 

*T. vaginalis* is a mucosal parasite, where adherence to host urogenital epithelial cells is critical for the initiation and maintenance of infection [12]. Upon contact with host cells, *T. vaginalis* showed strong upregulation of actin and actin-associated genes among other major transcriptomic changes [19], suggesting that the cytoskeletal transformation of *T. vaginalis* is important for its adherence [18,20,21]. Indeed, host cell adhesion was accompanied by significant morphology changes in these trophozoites [18,19]. Specifically, actively swimming *T. vaginalis* tend to be ellipsoidal, ovoidal, or even spherical in shape, while those incubated with human cells demonstrated amoeboid morphology and attached to their hosts via pseudopodia-like extensions [30]. This actin-based machinery was reported to also mediate *T. vaginalis* migration across host tissue [18]. ROS produced NADPH oxidases have been shown to act as critical regulators of the actin skeleton and cytoskeleton-supported cell functions [22,23,24]. Moreover, many proteins involved in cytoskeletal reorganization, including actin, GTPases, and integrins, were shown to be regulated in a redox-dependent manner [22,23,24]. Processes and molecules in *T. vaginalis* targeted by DPI, including nitric oxidase synthase and other flavin-dependent enzymes, may be similarly involved in the reorganization of the actin cytoskeleton during *T. vaginalis* adherence to host cells. 

Among the forefront of global health concerns is the emergence of drug-resistant pathogens. Currently, nitroimidazoles are the only class of antimicrobial drugs recommended for trichomoniasis treatment, and although highly effective, present significant vulnerability to emerging resistance due to the lack of alternative therapies. Metronidazole resistance is already being reported in 5–10% of clinical isolates [31,32]. Therefore, there is a pressing need for novel anti-*T. vaginalis* drugs. Based on our findings, DPI may be one such compound by inhibiting parasitic adhesion to host cells. Interestingly, a previous study similarly identified DPI as a nonantibiotic inhibitor exhibiting potent antimicrobial activity against drug-resistant strains of *Staphylococcus aureus* and *Mycobacterium tuberculosis* [33]. DPI has also been reported to be microbicidal against parasites of the genus *Leishmania* and *Trypanosoma* [34], as well as the malaria parasite Plasmodium falciparum [35]. Our data show that DPI does not have detrimental effects on host cells, suggesting that DPI and similar compounds may present an effective alternative therapy for trichomoniasis.

DPI acts by binding to and targeting flavin-containing oxidase enzymes, many of which reside within the mitochondria. For example, DPI was shown to inhibit the reduction of iron–sulfur clusters in mitochondrial NADH-ubiquinone oxidoreductase [36]. *T. vaginalis* expresses several iron–sulfur flavoprotein homologs in its mitochondrial-related organelles, called hydrogenosomes, that may act as possible targets of DPI [37,38]. The function of these iron–sulfur flavoprotein homologs is unknown but may be involved in electron transfer, which would allow DPI to selectively deplete *T. vaginalis* intracellular ATP levels. Given that actin reorganization and general cytoadherence require a large consumption of ATP, this may be responsible for *T. vaginalis* detachment [39]. Interestingly, DPI did not induce *T. vaginalis* cell death in our study. As such, DPI treatment may induce a viable but nonculturable (VBNC) state in *T. vaginalis*. DPI has been similarly reported to induce the rapid transition from an active to a VBNC state in *Mycobacterium tuberculosis* [40]. Collectively, these data strongly suggest that DPI contribute to parasite detachment from host cells during *T. vaginalis* infection. Further studies are needed to determine the exact mechanism(s) by which DPI leads to *T. vaginalis* detachment from SiHa cells.

## Figures and Tables

**Figure 1 microorganisms-08-01570-f001:**
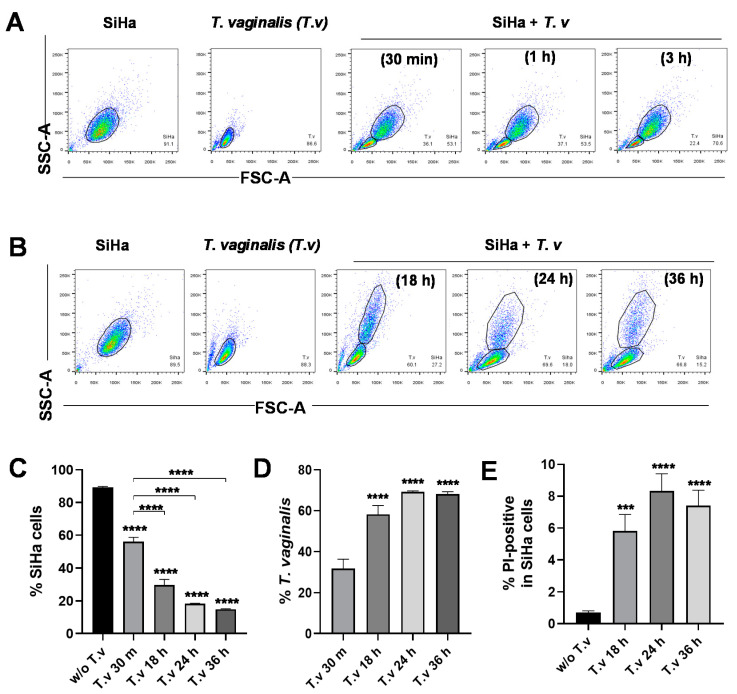
Flow cytometric analysis of parasite–host cell mixed culture. (**A**,**B**) SiHa cells were infected with live *T. vaginalis* (T.v) trophozoites for the indicated durations. Cells were then trypsinized and analyzed by flow cytometry. Flow cytometry data plots based on cell size (SSC-A and FSC-A) were used to identify *T. vaginalis* (small) and SiHa (large) cell populations. (**C**) SiHa cell levels were quantified from flow cytometry data (*n* = 3). (**D**) Bound *T. vaginalis* levels were quantified from flow cytometry data (*n* = 3). (**E**) Propidium iodide (PI) positive (+) dead SiHa cells at the time indicated (*n* = 3). Statistical difference by one-way ANOVA compared to no infection controls or 30 min infection. Mean ± SD, *** *p* < 0.001, **** *p* < 0.0001. Data are representative of at least three independent experiments.

**Figure 2 microorganisms-08-01570-f002:**
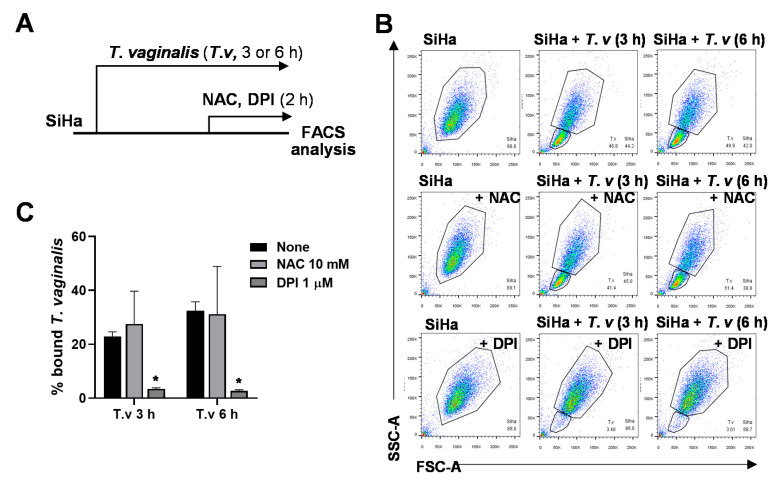
Antioxidant diphenyleneiodonium (DPI) reduced adherence of *T. vaginalis* to SiHa cells. (**A**) Experimental scheme: SiHa cells were infected with *T. vaginalis* for 3 or 6 h and treated with 10 mM NAC and 1 μM DPI for the last 2 h. Attached cells were analyzed by flow cytometry. (**B**) Representative flow cytometry profile. (**C**) Bound *T. vaginalis* levels were quantified from flow cytometry data (*n* = 3). Statistical difference by one-way ANOVA compared with no treatment controls (None). Mean ± SD, * *p* < 0.05. Data are representative of at least three independent experiments.

**Figure 3 microorganisms-08-01570-f003:**
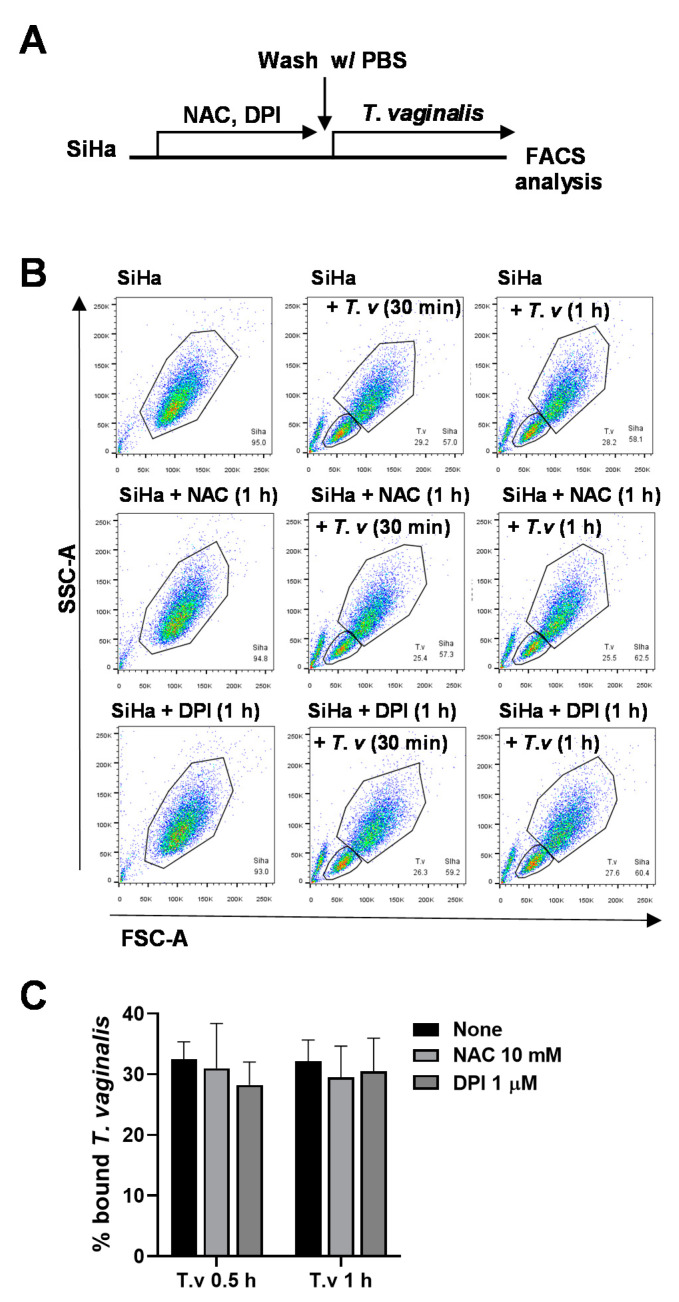
Pretreating SiHa cells with DPI did not affect *T. vaginalis* adherence. (**A**) Experimental scheme: SiHa cells were pretreated with 10 mM NAC and 1 μM DPI for 1 h. After washing with PBS, pretreated cells were infected with *T. vaginalis* for 30 min or 1 h and subjected to flow cytometry data analysis. (**B**) FSC-A and SSC-A dot plots for SiHa cells and *T. vaginalis* populations. (**C**) Bound *T. vaginalis* levels were quantified from flow cytometry data (*n* = 3). Mean ± SD, one-way ANOVA.

**Figure 4 microorganisms-08-01570-f004:**
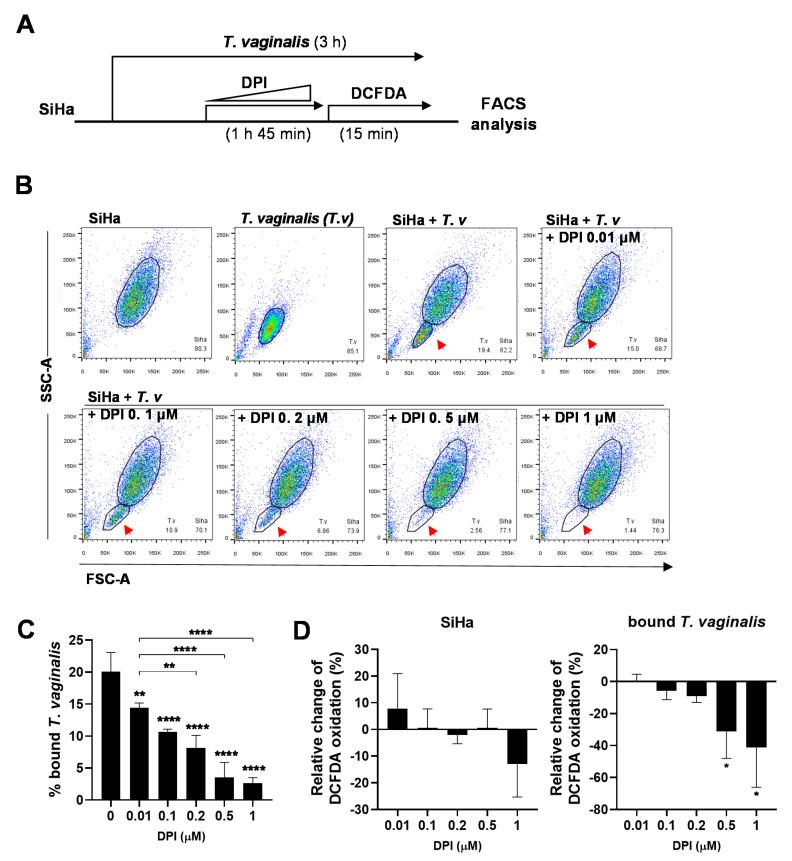
DPI induces detachment of *T. vaginalis* from SiHa cells in a dose-dependent manner and reduces ROS generation. (**A**) Experimental scheme: SiHa cells were infected with *T. vaginalis* for three hours, with DPI treatment at the indicated concentrations for the last 2 h prior to incubating with 1 μM CM-H_2_DCFDA for the last 15 min. Attached cells were analyzed by flow cytometry. (**B**) Representative FSC-A/SSC-A dot plot of the attached cell populations. Attached *T. vaginalis* cells are indicated by red arrows. (**C**) Bound *T. vaginalis* levels were quantified from flow cytometry data (*n* ≥ 3). (**D**) Levels of CM-H_2_DCFDA oxidation in SiHa cells and bound *T. vaginalis* relative to DPI untreated samples. Mean ± SD, one-way ANOVA. * *p* < 0.05, ** *p* < 0.01, **** *p* < 0.0001. The data represent triplicate experiments.

**Figure 5 microorganisms-08-01570-f005:**
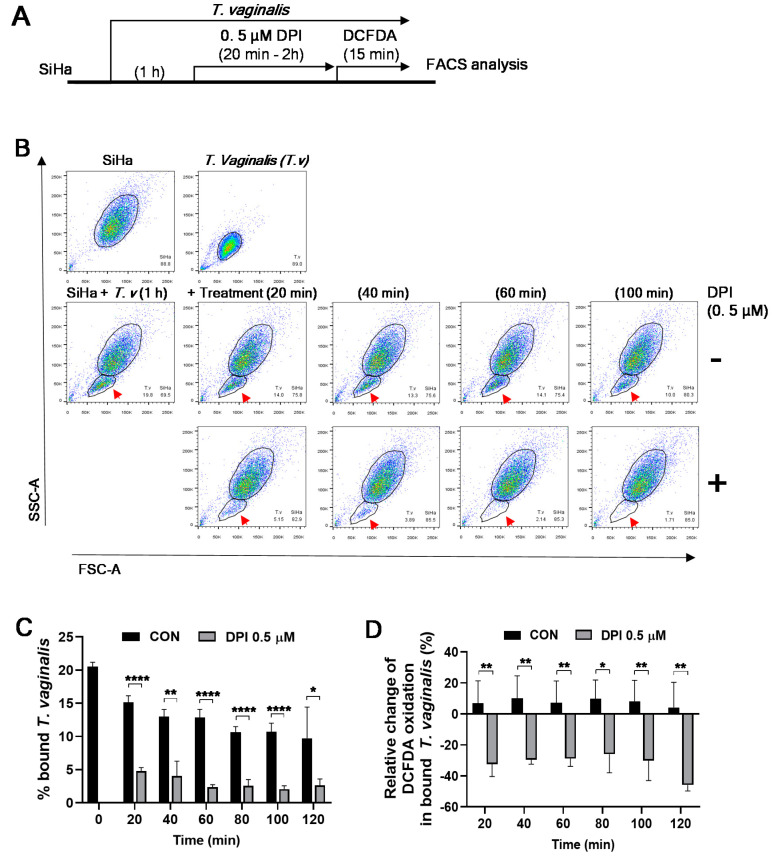
Kinetics of DPI-induced detachment of *T. vaginalis* from SiHa cells. (**A**) Experimental scheme: SiHa cells were infected with *T. vaginalis* for 1 h and treated with 0.5 μM DPI for the indicated durations. The mixed culture was incubated with 1 μM CM-H_2_DCFDA prior to trypsinization and flow cytometric analysis. (**B**) The FSC-A/SSC-A dot plot of SiHa cells and *T. vaginalis*. Attached *T. vaginalis* populations are indicated by red arrows. (**C**) Bound *T. vaginalis* levels were quantified from flow cytometry data (*n* ≥ 3). (**D**) Relative change of CM-H_2_DCFDA oxidation in the bound *T. vaginalis*. Mean ± SD. * *p* < 0.05, ** *p* < 0.01, **** *p* < 0.0001. Statistical difference by unpaired *t*-test compared to no treatment controls (CON). Data are representative of at least three independent experiments.

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
