# Peer review of "Adherence of Trichomonas vaginalis to SiHa Cells is Inhibited by Diphenyleneiodonium"

_microorganisms, 2020, doi:10.3390/microorganisms8101570_

Round 1

Reviewer 1 Report

This manuscript by Kim et al. describes the establishment of a flow cytometry-based method to examine the properties of T. vaginalis adhesion on the cervix carcinoma cell line, SiHa cells. They show that NADPH oxidase inhibitor diphenyleneiodonium (DPI) is involved in blockade of parasite adherence to SiHa cells in vitro. The manuscript is generally well written. There are just a few issues that require addressing:

  1. The authors need to provide representative microscopic images of SiHa cells in culture under the various treatments and infections used in the experiments.
  2. Based on the data they have presented, it is clear that DPI’s mode of action in inhibiting vaginalis binding to SiHa cells is not dependent on ROS reduction because the well known ROS scavenger (NAC) had no effect. For this reason, throughout the manuscript, the authors need to state clearly in their descriptions and discussion of results that targeting ROS reduction is not what led to inhibition of T. vaginalis binding. They should, therefore, suggest other modes of action.
  3. Line 118: add “than” after “greater”
  4. Line 235: correct phrase should read “may be involved” and not “may involve”.

Author Response

Hello. Please see the attachment.

Reviewer 2 Report

In this manuscript, Kim et al. find that the NADPH oxidase inhibitor DPI can inhibit the attachment of T. vaginalis to SiHa cells. Whilst the study is fairly straight forward, some edits are required before publication. In particular, the authors should explain where SiHa cells were chosen and why they are an appropriate cell line for this study. It seems a very strange choice of cell line to use and it would have been much better to use primary vaginal epithelial cells, which are widely available.

  • Ref 3 is not appropriate on line 36 - primary research showing these phenomena should be cited.
  • Line 47 - 'virulence' should be 'virulent'
  • Figure 1 - PI does not only identify apoptotic cells, but also necrotic cells. This should be make clear. 
  • Line 132 - reference needed.
  • The authors refer to their flow based analysis as 'FACS data'. This is incorrect as the authors performed no FACS analysis.
  • Figure 4 - why did DPI no result in a dose dependent decrease in CMH2DCFDA oxidation? This should be discussed.
  • Figure 5 - the authors state that detachment occurs 20-40mins after treatment. The 40 min flow panel should be shown as this is clearly an important one.

Author Response

Hello. Please see the attachment.
